# Role of Dysregulated Ion Channels in Sensory Neurons in Chronic Kidney Disease-Associated Pruritus

**DOI:** 10.3390/medicines6040110

**Published:** 2019-11-13

**Authors:** Akishi Momose, Micihihiro Yabe, Shigetoshi Chiba, Kenjirou Kumakawa, Yasuo Shiraiwa, Hiroki Mizukami

**Affiliations:** 1Department of Urology, Jusendo General Hospital, Koriyama 9638585, Japan; m.yabe@jusendo.or.jp (M.Y.); s.chiba@jusendo.or.jp (S.C.); k.kumakawa@jusendo.or.jp (K.K.); y.shiraiwa@jusendo.or.jp (Y.S.); 2Department of Pathology and Molecular Medicine, Hirosaki University Graduate School of Medicine, Hirosaki 0368563, Japan

**Keywords:** uremic pruritus, ion channels, cell signaling, Cav3.2 calcium channel, RT-PCR, skin

## Abstract

**Background:** We investigated ion channels at the skin, including peripheral nerve endings, which serve as output machines and molecular integrators of many pruritic inputs mainly received by multiple G protein-coupled receptors (GPCRs). **Methods:** Based on the level of chronic kidney disease–associated pruritus (CKD-aP), subjects were divided into two groups: non-CKD-aP (no or slight pruritus; n = 12) and CKD-aP (mild, moderate, or severe pruritus; n = 11). Skin samples were obtained from the forearm or elbow during operations on arteriovenous fistulas. We measured ion channels expressed at the skin, including peripheral nerve endings by RT-PCR: Nav1.8, Kv1.4, Cav2.2, Cav3.2, BK_Ca_, Anoctamin1, TRPV1, TRPA1, and ASIC. **Results:** Expression of Cav3.2, BK_Ca_, and anoctamin1 was significantly elevated in patients with CKD-aP. On the other hand, expression of TRPV1 was significantly reduced in these patients. We observed no significant difference in the levels of Cav2.2 or ASIC between subjects with and without CKD-aP. TRPA1, Nav1.8, and Kv1.4 were not expressed. **Conclusions:** It was concluded that this greater difference in the expression of ion channels in the skin tissue including, specially cutaneous peripheral nerve endings in CKD patients with CKD-aP may increase generator potential related to itching.

## 1. Introduction

Itching can be caused by various stimuli, including mechanical, electrical, and chemical stimuli. Exogenous and endogenous chemical stimuli include amines, proteases, neuropeptides, inflammatory mediators, and certain drugs. However, in previous studies that compared immunostaining skin sections treated with various common pruritogens (histamine, acetylcholine (Ach), etc.), uremic substances, and other substances causing itching (e.g., β-endorphin and endothelin-1), no significant difference was observed between patients with and without CKD-aP, even though the immunostaining epidermis sections were stained regardless of CKD-aP [1].

Among uremic substances, for example, endothelin-1, which is classified of moderate molecular weight, and para-cresyl sulfate, a protein-bound uremic toxin, were reported as candidate substances that cause itching in patients with chronic kidney disease (CKD)-associated pruritus (CKD-aP) [2,3]. It is reported that uremic toxins stimulate the production of reactive oxygen species (ROS) and ROS act on ion channels which are associated with pruritus [4,5,6,7]. Although CKD patients undergo hemodialysis with a high-performance membrane or on-line hemodiafiltration, as well as long-term or frequent hemodialysis, to remove these uremic substances, approximately 40% of patients still suffer from CKD-aP [8].

These findings suggested that itching can be caused by receptors and other signals, instead of pruritogens. Therefore, we first examined the downstream ion channels because it is difficult to measure the complex and many downstream signals from ligands (pruritogens). Expressed ion channels in cutaneous peripheral nerve endings in healthy individuals are as follows: among the voltage-gated Na^+^ channels (Nav), Nav1.7, Nav1.8, and Nav1.9; among the voltage-gated K^+^ channels, Kv1.4, Kv3.4, and Kv7.2 (Kv); among the Ca^2+^-activated K^+^ channels (K_Ca_), big conductance (BK_Ca_); among the two-pore K^+^ channels, TREK1 and TRAAK; among the voltage-gated Ca^+^ channels (Cav), N-type (Cav2.2) and T-type (Cav3.2); among the Cl channels, calcium-activated chloride channel Anoctamin1 (TMEM16A); among the transient receptor potential (TRP) channels, TRP vanilloid 1 (TRPV1) and TRP ankyrin 1 (TRPA1); and acid-sensing ion channels (ASIC1) [9,10]. Among these channels, Nav1.8, Kv1.4, BK_Ca_, Cav2.2, and Cav3.2 are expressed only in the cutaneous peripheral nerve endings, whereas TRPV1, TRPA1, Anoctamin1, BK_Ca_, and ASIC are expressed not only in the peripheral nerve cells but also in keratinocytes at the skin.

Our goal was to test the hypothesis that many pruritogens including uremic toxins act on ion channels on the peripheral nerve endings directly or sensitize those via GPCRs and ROS indirectly, changing the output of multiple ion channels and increase generator potential and generate action potential related to CKD-aP, resulting in encoding the amplitude, frequency and quality of impulse of the peripheral nerve. To explore the peripheral neuronal mechanisms underlying itching, we compared the expression of several ion channels at the skin, including peripheral nerve endings between CKD patients with and without CKD-aP.

## 2. Materials and Methods

### 2.1. Subjects

This cohort-sectional study was approved by the Ethics Committee of Jusendo General Hospital. (Approval code: No.126). Each patient gave written informed consent. Between February 2016 and September 2018, we performed arteriovenous fistula surgery in 77 patients with CKD stage 5, either before the induction of hemodialysis (HD) or within 1 month after the induction of HD. Twenty-three patients (8 females and 15 males, 38–86 years old) and one control agreed to participate in the study. The control was a patient undergoing an operation for prostatic hypertrophy, who reported no pruritus. Patients with concomitant dermatitis, e.g., atopic dermatitis or psoriasis, were excluded. The degree of pruritus was determined according to Shiratori’s Japanese classification of itching (Table 1). When the degree of pruritus was different between daytime and nighttime, a larger value of score was defined as the pruritus score. Based on the level of pruritus, subjects were divided into two groups: non-CKD-aP (no or slight pruritus; n = 12) and CKD-aP (mild, moderate, or severe pruritus; n = 11).

### 2.2. Study Designs

We compared age, gender, underlying disease, presence of hepatitis B or C, number of patients receiving treatment for itching, as well as levels of blood serum albumin, corrected Ca, inorganic phosphorus, intact parathyroid hormone (intact-PTH), blood serum hypersensitive C-reactive protein (hs-CRP), and blood ferritin in patients of the pruritus and non-pruritus groups.

Skin samples about 10 × 5 mm^2^ were obtained from the forearm or elbow immediate after commencing operations on the arteriovenous fistulae.

### 2.3. Quantitative Real-Time Polymerase Chain Reaction (RT-PCR)

Expressed ion channels in cutaneous peripheral nerve endings in healthy individuals are as follows: among the voltage-gated Na^+^ channels (Nav), Nav1.7, Nav1.8, and Nav1.9; among the voltage-gated K+ channels, Kv1.4, Kv3.4, and Kv7.2 (Kv); among the Ca^2+^-activated K^+^ channels (K_Ca_), big conductance (BK_Ca_); among the two-pore K+ channels, TREK1 and TRAAK; among the voltage-gated Ca^+^ channels (Cav), N-type (Cav2.2) and T-type (Cav3.2); among the Cl channels, calcium-activated chloride channel Anoctamin1 (TMEM16A); among the transient receptor potential (TRP) channels, TRP vanilloid 1 (TRPV1) and TRP ankyrin 1 (TRPA1); and acid-sensing ion channels (ASIC1) [5,6]. Among the ion channels expressed in cutaneous peripheral nerve endings, we measured the levels of Nav1.8 (SCN10A), Kv1.4 (KCNA4), BK_Ca_ (KCNMA1), Cav2.2 (CACNA 1B), Cav3.2 (CACNA 1H), Anoctamin1 (TREM16A), TRPV1, TRPA1, and ASIC by RT-PCR. Nav1.8, Kv1.4, Cav2.2, and Cav3.2 are expressed only in the cutaneous peripheral nerve endings, whereas TREM16A, TRPV1, TRPA1, ASIC1, and BK_Ca_ are expressed not only in the peripheral nerve cells but also in keratinocytes at the skin.

Gene expression levels for ion channels were determined after reverse transcription of RNA samples by quantitative PCR by using an ABI PRISM 7000 Sequence Detector (Thermo Fisher Scientific, Waltham, MA, USA) as described previously [11]. Pre-made TaqMan^®^ Gene Expression Assays for human were used (Table 2). Total RNA was isolated from the biopsied skin specimens with TRlzol (Thermo Fisher Scientific, Waltham, MA, USA). cDNA was synthesized from total RNA with reverse transcriptase reaction using SuperScript^TM^ VILO^TM^ Naster Mix (Thermo Fisher Scientific) following the manufacturer’s protocol. cDNA was synthesized from 1 μg of total RNA. For standardization of quantitation, beta 2 microglobulin as amplified simultaneously. The expression level of each gene is presented as fold increase in the subjects showing itchy compared with control subject.

### 2.4. Statistical Analysis

All values are expressed as means ± standard deviations (SD) for normally distributed data, medians (ranges) for non-normally distributed data, and numbers (percentages). No data points were excluded. All statistical analyses were performed using the unpaired *t*-test or m × n chi-square test for categorical outcomes. *p*-values < 0.05 were considered to be statistically significant.

## 3. Results

### 3.1. Baseline Characteristics

No differences were observed between the two groups in term of age, gender, underlying disease, duration of renal replacement therapy, presence of hepatitis B or C, number of patients receiving treatment for itching, or the levels of corrected Ca, inorganic phosphorus, blood serum albumin, blood serum hsCRP, and blood serum ferritin (Table 3).

### 3.2. Analysis of Ion Channels at the Skin by RT-PCR

Expression of Cav3.2, BK_Ca_, and Anoctamin1 was significantly higher in patients with CKD-aP. On the other hand, expression of TRPV1 was significantly decreased in those with CKD-aP. No significant difference in Cav2.2 and ASIC was observed between groups. TRPA1, Nav1.8, or Kv1.4 were not expressed (Table 4).

## 4. Discussion

The primary receptors responsible for itch are G protein-coupled receptors (GPCRs) (approximately 800 types in human), which then transmit itch signals to trimeric proteins, effectors, second messengers, as well as targets [12]. These transmitted signals ultimately affect the expression or function (i.e., sensitization) of ion channels [13,14]. These ion channels depolarize receptor potentials, and action potentials occur when the total depolarized receptor potential exceeds the threshold of voltage-dependent Na+ channels. Consequently, peripheral nerve signals are encoded. These signals (impulses) are transmitted to the central nervous system, including the spinal cord and brain, resulting in itching and scratching. Therefore, ion channels seem to act as output machines and molecular integrators of many pruritic inputs, which are mainly received by multiple GPCRs.

The Cav3.2 T-type calcium channel, which is only expressed in peripheral nerves at the skin, is associated with depolarization (i.e., receptor potential), action potential generation, and itching [15]. Our study demonstrated that the expression of Cav3.2 T-type calcium channels was significantly higher in patients with CKD-aP compared to those without CKD-aP. Therefore, it was speculated that the threshold of itch in patients with CKD-aP decreased because the expression of Cav3.2 T-type calcium channels in the peripheral nerve endings increased. Many substances causing itching, such as histamine, up-regulate the Cav3.2 T-type Ca^2+^ channel via GPCR [16,17]. For example, a previous study reported that agonists of the ACh muscarinic receptor, which is a GPCR, increases T-currents in rats. This report suggested that the Cav3.2 T-type Ca^2+^ channel is associated with itching in patients with atopic disease who received ACh intradermally [18]. By contrast, another study reported that Cav3.2 T-type calcium channels are more sensitive to inhibition by metals, such as zinc, copper, and nickel [19]. Furthermore, the major natural and mammalian endogenous fatty acids, including γ-linolenic acid and arachidonic acid, as well as the fully polyunsaturated ω3-fatty acids that are enriched in fish oil are potent inhibitors of the Cav3.2 T-type calcium channels [20]. This inhibitory effect may allow some patients with CKD-aP to benefit from treatment with zinc, γ-linolenic acid (ω-3), and fully polyunsaturated ω3-fatty acids [21].

Ca3.2 T-channels regulate cellular excitability in the peripheral nerve endings of nociceptors, whereas Cav2.2 (N-type Ca^2+^ channels), which also is expressed in peripheral nerves, regulate the release of neurotransmitters such as glutamate and substance P in the central terminals of nociceptor neurons in the spinal dorsal horn [22]. No statistically significant difference was observed in expression of Cav2.2 calcium channels between patients with and without CKD-aP. These findings suggest that Cav2.2 calcium channels are mainly expressed on the central side of the peripheral nerve, not at ending side, and can be involved in the transmission pathway of the itching.

Nav1.8 contributes to action potential generation in dorsal root ganglion (DRG) neurons in mice. Also, voltage-gated potassium (Kv) channels shape action potentials by controlling the repolarization phase, and they also determine the membrane potential and duration of the inter-spike interval [23]. Based on these findings, we measured Nav1.8 and Kv1.4, which are also expressed only expressed in peripheral nerves at the skin [24], but neither Nav1.8 nor Kv1.4 were expressed in the cutaneous peripheral nerve ending in CKD patients. This suggested that Nav1.7 (PN1), Nav1.9 (PN5), or Kv7.1-Kv7.5 (KCNQ channels) may be more associated with CKD-aP.

TRPV1 are non-selective cation channels that act as biosensors for environmental and noxious stimuli, as well as changes in temperature and conditions inside the cell. In addition to capsaicin and resiniferatoxin (RTX), protons (pH < 5.7), heat (> 42 °C), and multiple other ligands (endogenous lipids and metabolic products of lipoxygenase) can directly activate TRPV1 [25,26] Furthermore, indirect activation of TRPV1 by pruritogens such as histamine appears to require an intracellular signal transduction mechanism that lies downstream of GPCRs (i.e., sensitization) [27]. TRPV1 expression was significantly lower in patients with CKD-aP. Because TRPV1 is expressed not only on sensory neurons, but also on keratinocytes and mast cells, reduced expression of TRPV1 in CKD patients with CKD-aP is not limited to the peripheral nerve ending. Given that cutaneous TRPV1 expression reflects its expression in cutaneous peripheral nerve endings, TRPV1 expression may be more associated with pain rather than itching, and indeed it may inhibit itching. Khomula et al. showed that the TRPV1 channels are down-regulated, and Cav3.2 T-type channels are up-regulated, under normalgesic types of peripheral diabetic neuropathy in streptozotocin-induced diabetes rats [28]. The pathophysiology of CKD-aP may be similar to that of peripheral diabetic neuropathy.

TRPA1, which is also expressed on cells other than peripheral nerve cells can be activated by various reactive compounds, including mustard oil, cinnamaldehyde, formalin, and hydrogen peroxide, as well as noxious cold temperature, reactive oxygen species (ROS), and inflammatory lipids (4-hydroxynonenol). Wilson et al. showed that TRPA1 is the downstream target of both mas-related GPCR (Mrgpr) A3 and Mrgprc11, which act as receptors for the pruritogens chloroquine and BAM8-22, respectively [29]. However, no TRPA1 expression was observed regardless of CKD-aP, suggesting that TRPA1 may be down-regulated by intracellular signals of itch such as H_2_S [30]. Uremic toxins stimulate the production of ROS, such as H_2_O_2_, in mitochondria [4,5]. In addition, ROS sensitize Cav3.2 T-type calcium channels and Na^+^ channels, as well as TRPA1 and TRPV, and also increase intracellular Ca^2+^ [6,7]. Thus, uremic toxins may be related to Cav3.2 T-type calcium channels through ROS as well as GPCRs.

In general, membrane depolarization increases the sensitivity of Anoctamin1 to an increase in intracellular Ca^2+^ [31]. Anoctamin1, which is also expressed not only on sensory neurons, but also on keratinocytes, further enhances the depolarization of receptor potentials by efflux of anion Cl^−^ from the cell [32]. Compared to patients without CKD-aP, Anoctamin1 expression increased in patients with CKD-aP. Given that cutaneous TRPV1 expression reflects its expression in cutaneous peripheral nerve endings, it was thought that sensitivity or expression of Anoctamin1 increased due to elevated intracellular Ca^2+^ via Cav3.2 T-type calcium channels.

There is evidence that members of the Cav3.2 T-type calcium channel can physically and functionally interact with the BK_Ca_ channels, which play a key role in controlling action potential repolarization [33]. BK_Ca_ expression, which is also expressed on cells other than peripheral nerve cells, was significantly increased in patients with CKD-aP due to the increase of intracellular Ca^2+^ through the Cav3.2 T-type calcium channel. If cutaneous BK_Ca_ expression reflects its expression in cutaneous peripheral nerve endings, this finding suggested that repolarization of action potentials, leading to itching, was activated; this might affect the shape or frequency of action potential impulses related to itching in cutaneous peripheral nerve ending. In fact, the frequency of itch impulses is less than that of pain impulses in peripheral c-fibers [34].

ASCI was higher in patients with CKD and H+ sensitivity than in healthy individuals, but the difference was not statistically significant, suggesting no direct impact of pH on CKD-aP.

Many times in the past a new treatment option has been reported to be effective, but very soon thereafter conflicting results appear. Most therapeutic trials have shown only limited success. We think that this is because pruritogens mainly act on ion channels indirectly through many GPCRs or ROS and do not act on ion channels directly such as on pain stimuli.

This cohort-sectional study had some limitations. First, the total number of samples were relatively small, which may restrict interpretation of our results. Due to the small number of subjects, it was divided into two groups (non-CKD-aP and CKD-aP) instead of five groups (no, slight, mild, moderate and severe pruritus group). However, this separating way may be less meaningful, because each pruritus has different molecular profile.

Second, we examined ion channels specifically expressed in peripheral nerve endings (e.g., Cav3.2) and those expressed in keratinocytes or other sites, as well as peripheral nerve ending (e.g., TRPV1). Presently, it is technically difficult to separate the peripheral nerves from skin cells and examine ion channels expressed only the peripheral nerve endings. This requires further research. We think that the combination of ion channel agonists and antagonists will lead to new drug discoveries in the future. In addition to expression studies, functional studies are also needed to investigate the physiological roles of these channels. There are many reports that increased membrane expression of the ion channel parallels functional up-regulation of the ion channel in neurons [35].

## 5. Conclusions

It was concluded that a greater difference was observed in expression of ion channels at the skin tissue including, specific, for cutaneous peripheral nerve endings in CKD patients with CKD-aP than in those without CKD-aP. This up-regulated expression of Cav3.2 T-type calcium channel of the peripheral nerve in patients with CKD-aP may increase the generator potential and induce action potentials related to itching.

## Figures and Tables

**Table 1 medicines-06-00110-t001:** Definitions of Shiratori’s itch severity scores.

Score (Severity)	Daytime Symptoms	Nighttime Symptoms
4 (severe)	Intolerable itching, worsened instead of relieved by scratching. Cannot focus on work or study	Can hardly sleep because of itching. Scratching all the time, but itching intensifies with scratching
3 (moderate)	Scratching even in the presence of others. Irritation as a result of itching, continuous scratching	Wake up because of itching. Can fall asleep again after scratching, but continue to scratch unconsciously while sleeping
2 (mild)	Itch sensation is relieved by light, occasional scratching. Not too disturbing	Feel somewhat itchy, which is relieved by scratching. Do not wake up because of itch sensations
1 (slight)	Feel itchy sometimes, but tolerable without scratching	Feel slightly itchy when going to sleep, but do not need to scratch. Sleeping well
0 (no symptoms)	Hardly feel itchy or do not feel itchy at all	Hardly feel itchy or do not feel itchy at all

**Table 2 medicines-06-00110-t002:** Gene Assays ID.

Gene Name	Assay ID
ASIC1(acid sensing ion channel subunit 1)	Hs00952807_m1
Anoctamin 1	Hs00216121_m1
CACNA 1B (Cav2.2)	Hs04996252_m1
CACNA1H (Cav3.2)	Hs01103527_m1
KCNA4 (Kv1.4)	Hs00937357_s1
KCNMA1 (BK_Ca_)	Hs01119504_m1
SCN10A (Nav1.8)	Hs01045151_m1
TRPA1	Hs00175798_m1
TRPV1	Hs00218912_m1
beta-2-microglobulin	Hs00187842_m1

Pre-made TaqMan^®^ Gene Expression Assays for human were used (Thermo Fisher Scientific, Waltham, MA, USA).

**Table 3 medicines-06-00110-t003:** Patient characteristics.

Characteristic	Non-Pruritus (n = 12)	Pruritus (n = 11)	*p* Value
Degree of pruritus	none, slight	mild, moderate, severe	
Gender (F/M)	4/8	4/7	>0.05
Age (y.o.)	68 ± 10	68 ± 10	>0.05
Original disease (DM/CGN/PCK/unknown)	7/1/2/2	9/1/0/1	>0.05
HBV/HCV (n)	0/1	0/0	>0.05
Duration of HD (days)	23 (0–12779)	9 (0–5318)	>0.05
Albumin (d/dL)	3.2 ± 0.6	3.3 ± 0.4	>0.05
Corrected Ca (mg/dL)	8.8 ± 1.1	8.9 ± 0.6	>0.05
iP (mg/dl)	5.0 ± 1.8	5.7 ± 1.3	>0.05
i-PTH (pg/mL)	219 ± 126	233 ± 135	>0.05
hsCRP (mg/dL)	0.17 (0.02–1.44)	0.32 (0.04–8.00)	>0.05
Ferritin (ng/mL)	119 ± 84	130 ± 59	>0.05
Anti-pruritic therapy (nalfurafine, urea, predonisolone, crotamiton, diphenhydramine)	7 (58%)	3 (27%)	>0.05

Values are presented as means ± SD, median (range), or numbers (percentages). *p*-values were calculated using the unpaired t-test, Mann-Whitney U-test, or chi-square test. DM, diabetes mellitus; CGN, chronic glomeluronephritis; PCK, polycystic kidney; hsCRP, hypersensitivity C-reactive protein.

**Table 4 medicines-06-00110-t004:** Relative expression levels.

Gene Name		Non-Pruritus	Pruritus	*p*-Value
Cav3.2	CACNA 1H	0.948 (0.660–1.809) (n = 6)	2.490 (0.910–4.993) (n = 6)	0.039
Cav2.2	CACNA 1B	1.344 (0.038–19.186) (n = 5)	0.089 (0.066–0.977) (n = 3)	>0.05
Anoctamin1	TMEM 16A	1.094 (0.653–1.517) (n = 10)	1.528 (0.819–6.733) (n = 11)	0.009
ASIC1		0.796 (0.505–3.000) (n = 8)	2.962 (0.334–14.189) (n = 7)	>0.05
Kv1.4	KCNA4	No date	No date	
Na1.8	SCN10A	No date	No date	
TRPA1		No date	No date	
TRPV1		1.013 (0.804–1.223) (n = 3)	0.394 (0.256–0.463) (n = 3)	0.048
KCa1.1 (BK_Ca_)	KCNMA1	0.911 (0.526–1.685) (n = 7)	2.657 (0.664–4.042) (n = 7)	0.020

Values are expressed as median (range). *p*-values were calculated using the unpaired *t*-test. BK_Ca_, large conductance in Ca^2+^-activated K^+^ channels.

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
