# Peer review of "Role of Dysregulated Ion Channels in Sensory Neurons in Chronic Kidney Disease-Associated Pruritus"

_medicines, 2019, doi:10.3390/medicines6040110_

Round 1

Reviewer 1 Report

The manuscript is very interesting, in fact it is still debated in the literature the pruritus in patients with CKD.

The results are also very interesting for clinical practice, considering that the itching affects 40% of the patients in dialisis, a population that continues to grow, especially in the elderly, which could significantly reduce the quality of life

However “….These findings suggest that Cav2.2 calcium channels can induce itching in central nerves, instead of 159 peripheral nerves…” this statement reported by the authors could be a bit risky

Also the conclusions of the manuscript also appear a bit risky

The authors should better write the pathogenetic hypotheses of itching in CKD in the introduction

The authors also talk about the involvement of substances such as endothelin 1 or uremic toxins derived from intestinal dysbiosis without explaining their precise function in pruritus. The topic should be explored even if synthetically.

However it is certainly a well designed and well written manuscript

Author Response

Response to Reviewer 1 Comments

Thank you very much for having reviewed our manuscript entitled “ Chronic kidney disease-associated pruritus is caused by dysregulated ion channels in sensory neurons”. We have changed our manuscript as you suggested. The details of the responses to each advise were written below.

Point 1: However “….These findings suggest that Cav2.2 calcium channels can induce itching in central nerves, instead of 159 peripheral nerves…” this statement reported by the authors could be a bit risky.

Response 1 (Line159): Thanks. Your comment is very appropriate.

#1 (Line 155) We deleted “only expressed”.

#2 (Line 156) We deleted “at the skin”

#3 (Line159) We changed from the following sentence: “….These findings suggest that Cav2.2 calcium channels can induce itching in central nerves, instead of 159 peripheral nerves…” to “These findings suggest that Cav2.2 calcium channels are mainly expressed at the central side of the peripheral nerve, not at ending side, and can be involved in the transmission pathway of the itching.

Point 2: Also the conclusions of the manuscript also appear a bit risky.

Response 2 (Line 227): Thanks. Your comment is very appropriate. We changed following sentence.

From “….This upregulated expression of Cav3.2 T-type calcium channel of the peripheral nerve in patients with CKD-aP may increase the generator potential related to itching and induce action potentials, resulting in encoding the amplitude, frequency and quality of impulses of the peripheral nerve neurons….” to “….This upregulated expression of Cav3.2 T-type calcium channel of the peripheral nerve in patients with CKD-aP may increase the generator potential and induce action potentials related to itching….”

Point 3: The authors should better write the pathogenetic hypotheses of itching in CKD in the introduction

Response 3 (Line54): Thanks. We changed following sentence.

From “Our goal was to test the hypothesis that the output of multiple ion channels is associated with itching or pain.” To “Our goal was to test the hypothesis that many pruritogens including uremic toxins act on ion channels on the peripheral nerve endings directly or sensitize those via GPCRs and ROS indirectly, changing the output of multiple ion channels and increase generator potential and generate action potential related to CKD-aP, resulting in encoding the amplitude, frequency and quality of impulse of the peripheral nerve.”

Point 4: The authors also talk about the involvement of substances such as endothelin 1 or uremic toxins derived from intestinal dysbiosis without explaining their precise function in pruritus. The topic should be explored even if synthetically.

Response 4 (Line38): Thanks. We added them as follows:

It is reported that uremic toxins stimulate the production of reactive oxygen species (ROS) and ROS act on ion channels which are associated with pruritus [4-7].

Reviewer 2 Report

Dear Authors,

Thank you for writing this excellent paper which adds knowledge to this slowly emerging area of the pathogenesis of uremic pruritus.

Correctly, in your limitations section (commencing Line 215) you acknowledge that, because some ion channels normally appear on both nerve afferents and skin cells (such as keratinocytes) that it is difficult to know in skin tissue samples whether the ion channels that are more expressed are on afferents or skin cells or both. You could state that this requires further research.

I have some small suggestions in the text :

Line 23. Could change this line to read : "in expression of ion channels in skin tissue including, specifically, for cutaneous peripheral nerve..." Line 38. Add "a" before high-performance so it reads "with a high-performance..." Line 129. You state " The primary receptors responsible are G protein..." Do you mean responsible for itch ?  If so, you could state  "The primary receptors responsible for itch are G protein..." Line 182. Add "on" so it reads "TRPA1, which is also expressed on cells other.." Line 202. Rather than "expressed cells" should be "expressed on cells" Line 212. Rather than "I" should be "we". There is more than one author. Line 213. Rather than "pruritogens mainly acts on ion channels" should read "pruritogens mainly act on ion channels" Line 213. End of the line. To change to read : "through many GPCRs or ROS and do not act..." Line 219. The "I think" should be "We think". Line 220. First word - should be "discoveries" Line 221. To change from "There are many studied reports" to "There are many reports" Line 226. change to "tissue including, specifically, for cutaneous..." Line 228. Insert "the" in front of "generator" so it reads "may increase the generator"

Author Response

Response to Reviewer 2 Comments

Thank you very much for having reviewed our manuscript entitled “ Chronic kidney disease-associated pruritus is caused by dysregulated ion channels in sensory neurons”. We have changed our manuscript as you suggested. The details of the responses to each advise were written below.

Point 1:Correctly, in your limitations section (commencing Line 215) you acknowledge that, because some ion channels normally appear on both nerve afferents and skin cells (such as keratinocytes) that it is difficult to know in skin tissue samples whether the ion channels that are more expressed are on afferents or skin cells or both. You could state that this requires further research.

Response 1 (Line215): Yes, your comment is surely appropriate, and, thus, added the following sentences in limitation section: Now it is technically difficult to separate the peripheral nerves from skin cells and examine ion channels expressed only the peripheral nerve endings.This requires further research.

Point 2: Line 23. Could change this line to read : "in expression of ion channels in skin tissue including, specifically, for cutaneous peripheral nerve..."

Response 2: Thanks. We revised them as you suggested.

Point 3: Line 38. Add "a" before high-performance so it reads "with a high-performance..."

Response 3: Thanks. We added “a” as you suggested.

Point 4: Line 129. You state " The primary receptors responsible are G protein..." Do you mean responsible for itch ?  If so, you could state  "The primary receptors responsible for itch are G protein..."

Response 4: Thanks. We revised them as you suggested.

Point 5: Line 182. Add "on" so it reads "TRPA1, which is also expressed on cells other.."

Response 5: Thanks. We revised them as you suggested.

Point 6: Line 202. Rather than "expressed cells" should be "expressed on cells" Response 6: Thanks. We revised them as you suggested.

Point 7: Line 212. Rather than "I" should be "we". There is more than one author.

Response 7: Thanks. We revised them as you suggested.

Point 8: Line 213. Rather than "pruritogens mainly acts on ion channels" should read "pruritogens mainly act on ion channels"

Response 8: Thanks. We revised them as you suggested.

Point 9: Line 213. End of the line. To change to read : "through many GPCRs or ROS and do not act..."

Response 9: Thanks. We revised them as you suggested.

Point 10: Line 219. The "I think" should be "We think". Line 220. First word - should be "discoveries"

Response 10: Thanks. We revised them as you suggested.

Point 11: Line 221. To change from "There are many studied reports" to "There are many reports"

Response 11: Thanks. We revised them as you suggested.

Point 12: Line 226. change to "tissue including, specifically, for cutaneous..."

Response 12: Thanks. We revised them as you suggested.

Point 13: Line 228. Insert "the" in front of "generator" so it reads "may increase the generator"

Response 13: Thanks. We inserted “the” as you suggested.
